# Future Designs of Clinical Trials in Nephrology: Integrating Methodological Innovation and Computational Power

**DOI:** 10.3390/s25164909

**Published:** 2025-08-08

**Authors:** Camillo Tancredi Strizzi, Francesco Pesce

**Affiliations:** 1Department of Translational Medicine and Surgery, Università Cattolica del Sacro Cuore, 00168 Rome, Italy; camillotancredi.strizzi01@icatt.it; 2Nephrology, Dialysis and Transplantation Unit, Fondazione Policlinico Universitario A. Gemelli IRCCS, 00168 Rome, Italy; 3Division of Renal Medicine, Ospedale Isola Tiberina—Gemelli Isola, 00186 Rome, Italy

**Keywords:** nephrology, clinical trial design, artificial intelligence (AI), in silico clinical trials (ISCT), computational modeling

## Abstract

**Highlights:**

**What are the main findings?**
Clinical research in nephrology faces persistent challenges that can be addressed by combining two key innovation streams: advanced trial methodologies (like adaptive and pragmatic designs) and powerful computational tools, including Artificial Intelligence (AI) and in silico clinical trials (ISCTs).Specific computational tools are emerging that may offer targeted solutions. For example, Augmented Reality (AR) shows promise for enhancing the precision of interventional procedures like biopsies, while Conditional Tabular Generative Adversarial Networks (CTGANs) are being investigated as a method to generate synthetic data to help address scarcity in rare disease research.

**What is the implication of the main finding?**
The synergistic integration of advanced trial designs with AI-driven analytics and in silico simulations has the potential to provide a clear pathway toward conducting clinical trials that are faster, more precise, more cost-effective, and better tailored to individual patient needs.Realizing this potential is contingent upon the nephrology community proactively addressing significant implementation barriers related to data quality, model validation, evolving regulatory standards, and ethical oversight.

**Abstract:**

Clinical trials in nephrology have historically been hindered by significant challenges, including slow disease progression, patient heterogeneity, and recruitment difficulties. While recent therapeutic breakthroughs have transformed care, they have also created a ‘paradox of success’ by lowering baseline event rates, further complicating traditional trial designs. We hypothesize that integrating innovative trial methodologies with advanced computational tools is essential for overcoming these hurdles and accelerating therapeutic development in kidney disease. This narrative review synthesizes the literature on persistent challenges in nephrology trials and explores methodological innovations. It investigates the transformative impact of computational tools, specifically Artificial Intelligence (AI), techniques like Augmented Reality (AR) and Conditional Tabular Generative Adversarial Networks (CTGANs), in silico clinical trials (ISCTs) and Digital Health Technologies across the research lifecycle. Key methodological innovations include adaptive designs, pragmatic trials, real-world evidence, and validated surrogate endpoints. AI offers transformative potential in optimizing trial design, accelerating patient stratification, and enabling complex data analysis, while AR can improve procedural accuracy, and CTGANs can augment scarce datasets. ISCTs provide complementary capabilities for simulating drug effects and optimizing designs using virtual patient cohorts. The future of clinical research in nephrology lies in the synergistic convergence of methodological and computational innovation. This integrated approach offers a pathway for conducting more efficient, precise, and patient-centric trials, provided that critical barriers related to data quality, model validation, regulatory acceptance, and ethical implementation are addressed.

## 1. Introduction

Clinical research in nephrology has navigated a complex trajectory. For many years, progress lagged significantly behind advancements in research on other chronic disease like cardiology and oncology, despite the profound burden imposed by kidney diseases in terms of morbidity, mortality, and healthcare expenditure [1]. The landscape, however, has been fundamentally reshaped over the past decade by a series of high-quality, large-scale randomized controlled trials (RCTs) that delivered major therapeutic breakthroughs. This success, however, while practice-changing, has introduced a new challenge for future clinical research, a ‘paradox of success’ [2]. The success of new therapies, such as SGLT2 inhibitors, effectively lowers the baseline event rate (e.g., progression to end-stage kidney disease (ESKD) or significant estimated glomerular filtration rate (eGFR) decline) in the patient populations recruited for new trials. Consequently, demonstrating the incremental benefit of a novel investigational agent requires a much larger sample size or a longer follow-up period to achieve the same statistical power, thereby increasing trial cost and complexity. This paradox compounds persistent hurdles that have long impeded progress, creating a dual challenge that demands a paradigm shift in trial design.

This paper synthesizes these evolving challenges and argues that the future of nephrology research depends on the synergistic integration of methodological innovations with advanced computational approaches. We conducted a narrative review of peer-reviewed literature published between 2015 and 2024 using PubMed, focusing on keywords ‘nephrology’, ‘adaptive trials’, ‘digital health’, and ‘in silico trials”. We outline the obstacles, analyze adaptive frameworks and new technologies, and provide a roadmap to accelerate the delivery of novel therapies to individuals affected by kidney disease worldwide (Figure 1).

This review uniquely integrates methodological and computational innovations specifically for nephrology trials, addressing a gap in the current literature that typically examines these approaches separately.

## 2. Persistent Challenges Shaping the Future of Trial Design

The impetus to rethink clinical trial design in nephrology stems from two interconnected sources: long-standing systemic barriers and a new statistical reality created by therapeutic success [1]. The slow progression of many kidney diseases, particularly chronic kidney disease (CKD), has always impacted trial feasibility due to low event rates for definitive outcomes like ESKD or mortality [2,3]. This necessitates prohibitively large and long trials, demanding substantial investment and delaying access to beneficial therapies. This statistical reality forces a critical evaluation of trial design, pushing towards strategies like focusing on high-risk enriched populations [4] or relying more heavily on validated surrogate endpoints [3,5], particularly for studying early-stage disease [6]. Simultaneously, inefficiencies in recruitment and retention represent a major operational barrier [7]. Failure to recruit representative populations, often due to narrow eligibility criteria, participant burden, and a lack of trust, severely compromises the generalizability and equity of research findings [8]. Historical reliance on narrow eligibility criteria often excluded patients seen in routine practice [9]. The consistent underrepresentation of minority and socioeconomically disadvantaged groups highlights deep-seated systemic issues that demand patient-centered designs and improved communication [7,8]. Furthermore, a “one-size-fits-all” approach to conditions as diverse as diabetic kidney disease, glomerulonephritis, genetic disorders, and acute kidney injury (AKI), risks diluting treatment effects and motivates the need for sophisticated stratification methods and flexible trial designs (Table 1).

## 3. Methodological Innovations: Trial Frameworks

In response to these challenges, a suite of methodological innovations is being adopted to make trials more flexible, efficient, and relevant. Adaptive trial designs represent a paradigm shift away from fixed protocols, allowing prospectively planned modifications based on accumulating interim data without compromising statistical integrity [10,11]. Common adaptations include sample size re-estimation, dropping ineffective treatment arms, or enriching the study population, which can lead to shorter, more efficient trials [2]. Platform trials, which use a master protocol to test multiple interventions concurrently against a shared control group, are a particularly powerful application of this, dramatically increasing efficiency for evaluating numerous therapies in heterogeneous diseases [12,13]. Recognizing the limitations of traditional RCTs in reflecting routine practice, there is a growing emphasis on pragmatic clinical trials (PCTs) and real-world evidence (RWE). PCTs evaluate interventions in real-world settings with broad eligibility criteria, increasing generalizability [14]. RWE, derived from sources like Electronic Health Records (EHRs) and registries, offers complementary insights into long-term effectiveness and safety in diverse populations [15,16]. These modern frameworks are increasingly powered by Digital Health Technologies (DHTs). Wearables, sensors, and apps enable remote monitoring and the collection of electronic patient-reported outcomes (ePROs), which can reduce participant burden and facilitate the development of more sensitive digital endpoints [17,18]. This facilitates the development of novel ‘digital endpoints,’ which differ from traditional clinical markers by capturing high frequency, real-world patient data. For instance, instead of relying solely on quarterly eGFR measurements, a digital endpoint might be a composite score derived from daily wearable data on physical activity, continuous blood pressure monitoring, and electronic patient-reported outcomes (ePROs), potentially detecting a patient’s clinical deterioration with greater sensitivity and speed. While traditional RWE from registries and claims databases provides valuable long-term insights, a new frontier lies in augmenting these sources with high-frequency sensor data, offering a richer, real-time view of patient health between clinic visits. This methodological shift is coupled with the drive to validate and accept surrogate endpoints, such as a significant decline in estimated eGFR slope [19]. While rigorous validation is ongoing, these markers are essential for making trials feasible when hard endpoints are rare.

## 4. Computational Transformation

### 4.1. The Role of Artificial Intelligence

Beyond adapting existing trial frameworks, the integration of computational power, particularly Artificial Intelligence (AI), offers a more profound transformation for clinical research in nephrology. By leveraging the ability of AI systems, including machine learning (ML), deep learning (DL), and natural language processing (NLP), to learn from vast datasets and identify complex patterns, researchers can address persistent trial challenges in novel ways across the entire research lifecycle (Figure 2) [20,21,22]. A key contribution of AI lies in optimizing trial design and predicting success before a study begins. Algorithms are being explored to analyze historical trial data, preclinical information, and real-world data with the goal of refining inclusion/exclusion criteria, selecting endpoints, and determining the optimal duration for new protocols [22]. Similarly, there is growing interest in using predictive models to estimate a trial’s probability of success, which could enable more rational resource allocation [22]. This optimization extends to pharmacokinetics and pharmacodynamics (PK/PD), where AI can model complex relationships to help refine dosing strategies more efficiently than traditional methods [23,24,25]. AI also offers revolutionary potential in accelerating patient identification and enabling precision stratification. NLP algorithms have the potential to rapidly screen large volumes of structured and unstructured EHR data, including clinical notes, to assist in identifying eligible participants, significantly reducing manual workload in proof-of-concept settings [26,27]. While NLP screens text-based records, this process can be synergistic with identifying patients who are most suitable for trials involving intensive, sensor-based remote monitoring. Following identification, ML models can integrate diverse patient data, clinical, demographic, genomic, and imaging, to perform sophisticated risk stratification. This allows for either prognostic enrichment (selecting high-risk patients to increase statistical power) or predictive enrichment (selecting patients most likely to respond to a specific intervention), offering a data-driven approach to managing the inherent heterogeneity of kidney disease populations [28]. During trial conduct and monitoring, AI enhances both efficiency and participant centricity. It is essential for interpreting the high-volume, continuous data streams generated by Digital Health Technologies, enabling near real-time remote monitoring [22,29]. Predictive models can identify patients at risk of non-adherence or dropout, allowing for proactive, targeted interventions, potentially supported by AI tools like chatbots [29]. AI has the potential to enhance Risk-Based Monitoring (RBM) by identifying anomalous data patterns or site issues and automating aspects of data management, although broad implementation is still in its early stages [22,27]. Finally, AI fundamentally enhances data analysis and insight generation, especially with the complex, multimodal datasets increasingly common in modern research. ML and DL algorithms excel at identifying intricate, non-linear relationships within high-dimensional data (e.g., multi-omics, RWD) that traditional statistical methods might miss, fueling predictive analytics for clinical outcomes like CKD progression or treatment response [20,21,30,31]. This capability is also indispensable for generating meaningful real-world evidence (RWE) from large, often messy data sources and allows for deeper scientific discovery by moving beyond hypothesis testing to data-driven hypothesis generation [15,22]. By moving beyond hypothesis testing to data-driven hypothesis generation, AI allows for deeper scientific discovery. Furthermore, ML algorithms are key to deriving potentially more sensitive or earlier “digital endpoints” from the continuous data collected by wearables and sensors, directly addressing the limitations of traditional endpoints. AI offers a promising toolkit for enhancing trial speed, precision, and efficiency by analyzing complex data in novel ways, although full integration into nephrology research workflows remains an emerging area [21].

### 4.2. Sensor Technology and Digital Endpoints

A cornerstone of the computational transformation in nephrology is the use of Digital Health Technologies (DHTs), specifically patient-centric sensors that generate continuous, real-world data streams [17]. These range from established tools like remote blood pressure cuffs and continuous glucose monitors (CGMs), critical for managing common comorbidities, to wearables tracking physical activity, sleep, and heart rate [32,33]. Emerging technologies such as bioimpedance sensors for hydration status and smart scales for daily weight measurement offer direct insights into key parameters for patients with CKD and ESKD [34]. The true power of AI in this context lies in sensor fusion, the ability to integrate and analyze signals from multiple, disparate devices. For example, an AI model could combine data from a smart scale (weight), a blood pressure cuff (hemodynamics), and a wearable accelerometer (activity levels) to create a predictive alert for an impending fluid overload event, offering a far more nuanced view than any single measurement could provide [35,36]. However, the development of sensor-derived ‘digital endpoints’ faces significant validation challenges. Rigorously proving that a novel digital marker, such as an algorithmically derived ‘frailty score’ from an accelerometer, reliably predicts clinical outcomes like hospitalization or mortality requires dedicated clinical studies [37]. Furthermore, regulatory agencies like the FDA and EMA are still developing clear frameworks for accepting device-derived data as primary or secondary evidence in pivotal trials, making early engagement and robust validation planning essential [38]. Real-time biosensor data is a natural fit for adaptive trial designs. For instance, an adaptive trial could use continuous data streams from sensors to monitor treatment response or adverse events in near real time [39]. Pre-defined rules in the protocol could allow for automated dose adjustments or patient allocation based on these incoming data streams, making the trial more dynamic and responsive than one relying on intermittent clinic visits.

### 4.3. Leveraging In Silico Clinical Trials

Complementing the data-driven analytical power of AI, the field of in silico clinical trials (ISCTs) offers a distinct yet synergistic computational approach focused on simulation [40]. ISCTs leverage computational modeling and simulation (CM&S) [41], a “third pillar” alongside in vitro and in vivo methods [42], to conduct virtual experiments [43]. The core innovation is the use of computationally generated “virtual patients” or “virtual cohorts” designed to represent the target patient population, including its inherent variability [43,44]. These virtual populations allow researchers to simulate interventions and predict outcomes in silico, providing powerful tools to address specific nephrology research challenges. One key innovative application in nephrology is predictive pharmacokinetic/pharmacodynamic (PK/PD) modeling, particularly using Physiologically Based Pharmacokinetic (PBPK) models [45]. Given that kidney function critically influences the elimination of many drugs, PBPK models provide a mechanism-based approach to simulate how varying degrees of CKD impact drug exposure [46]. This allows for a priori dosage adjustments for renally cleared drugs, potentially reducing the need for dedicated, resource-intensive renal impairment studies in humans. Specific PBPK models have been developed and applied to drugs relevant to kidney disease patients [47,48]. The use of virtual patient cohorts is another transformative aspect. There is growing interest in using these computationally constructed populations to augment traditional trials, for instance, by exploring the generation of “synthetic control arms” [43]. This approach holds promise for reducing the number of participants required in placebo or standard-of-care groups, which is especially valuable in rare kidney diseases (e.g., specific glomerulonephritis, Fabry disease) where patient recruitment poses a significant barrier [49]. Furthermore, virtual cohorts allow researchers to explore treatment effects in specific subpopulations or simulate scenarios, such as testing interventions under conditions of extreme physiological stress, that would be impractical or unethical to conduct in living subjects [50]. Beyond direct trial simulation, CM&S techniques underpin deeper exploration of pathophysiology and hypothesis testing [51]. Mechanistic models, including those used in Quantitative Systems Pharmacology (QSP), integrate existing biological knowledge to simulate complex disease processes [52]. This offers a promising approach for a quantitative framework to investigate the interplay of factors in conditions like CKD mineral and bone disorder (CKD-MBD), explore mechanisms in Fabry disease, simulate renal autoregulation, and test mechanistic hypotheses about drug action, moving beyond empirical observation towards a more system-level understanding [51,52]. ISCTs also provide a powerful platform for optimizing clinical trial design before initiation [42,44]. By simulating trials with varying parameters, inclusion/exclusion criteria, sample sizes, endpoint definitions, durations, researchers can computationally assess the likelihood of success and identify the most efficient and robust design. This predictive capability allows for more informed decisions, potentially saving significant time and resources by avoiding suboptimal designs. A crucial driver for ISCT adoption is their alignment with the ethical and efficiency goals embodied in the Reduce, Refine, Replace (RRR) principles for animal and human testing [42]. By providing human-relevant predictions computationally, ISCTs can potentially decrease reliance on animal models, which may have limited translational value, and reduce the number of human participants needed in physical trials. These diverse applications are increasingly recognized within the Model-Informed Drug Development (MIDD) paradigm, a strategy endorsed by regulatory agencies like the FDA and EMA [53]. MIDD encourages the systematic use of modeling and simulation throughout the drug development lifecycle to improve decision-making and efficiency.

### 4.4. Generative Adversarial Networks

Addressing the profound challenge of data scarcity in clinical trials for rare kidney diseases, which chronically suffer from small patient cohorts and inherent data heterogeneity, AI-driven data augmentation techniques offer transformative potential. Conditional Tabular Generative Adversarial Networks (CTGANs) are specifically engineered to learn the complex underlying distributions and intricate correlations within real-world tabular clinical data. Subsequently, CTGANs can generate high-fidelity synthetic patient records that statistically mirror the original dataset, effectively expanding limited datasets while aiming to preserve data privacy, as synthetic records do not correspond to actual individuals [54,55]. By augmenting small cohorts, CTGANs can improve the robustness of predictive models, increase statistical power for hypothesis testing, and enable more reliable subgroup analyses that would otherwise be infeasible [56]. For instance, synthetic data can be employed to balance arms in a trial, pre-train machine learning models for outcome prediction or patient stratification, or even contribute to the simulation of control arms, complementing ISCT methodologies [55]. This allows for more efficient utilization of scarce real patient data and can accelerate the evaluation of novel therapies. Nevertheless, the integration of CTGAN-generated data into the clinical trial ecosystem demands rigorous validation to ensure the synthetic data’s fidelity, utility, and privacy-preserving characteristics [57]. Critically, as these models learn from the initial, often small and potentially unrepresentative datasets typical of rare diseases, there is a risk of perpetuating or even amplifying existing biases if not carefully managed. The evolving regulatory landscape for synthetic data also underscores the need for transparent reporting and thorough validation to gain broader acceptance. The imperative for high-quality input data for CTGANs may also indirectly foster initiatives towards more standardized and comprehensive data collection protocols for rare kidney diseases, creating a positive feedback loop where the availability of advanced augmentation tools encourages better foundational data practices.

### 4.5. Augmented Reality

Further extending the repertoire of computational tools, Augmented Reality (AR) is emerging to enhance the precision and efficacy of interventional procedures pivotal to nephrology, with direct implications for clinical trial data quality and participant outcomes. By overlaying real-time, three-dimensional (3D) pre-operative imaging data (e.g., CT, MRI) or dynamic physiological information onto the clinician’s view of the operative field, AR systems provide intuitive guidance, addressing inherent limitations of traditional techniques [58]. In the critical area of hemodialysis vascular access, AR guidance has shown potential for improving cannulation accuracy and procedural efficiency, with studies indicating faster task completion and superior ergonomics for expert users employing HoloLens-based applications [59]. For image-guided renal biopsies, AR systems that fuse preoperative CT scans with intraoperative ultrasound enable more precise needle targeting, thereby enhancing diagnostic yield [60]. The integration of AR into such nephrology interventions can significantly benefit future clinical trial designs by standardizing complex procedures, thereby reducing inter-operator variability, a notable confounder in surgical trials. This enhanced procedural consistency is intended to promote more homogenous intervention delivery, which, in turn, could lead to higher quality, more reliable endpoint data. Moreover, AR facilitates the collection of novel, objective intraoperative data points (e.g., precise lesion localization, real-time instrument tracking) that could serve as richer, more granular endpoints or process measures, and its role in surgical training can ensure higher proficiency among trial personnel, bolstering trial integrity. However, the adoption of AR in trials also necessitates careful consideration of equitable technology access across sites and robust ethical frameworks for the use of AR-generated patient data.

### 4.6. Illustrating Synergy: AI-Enhanced Adaptive Trial in Action

The true power of these innovations is unlocked when they are integrated (Table 2). The process would begin by using Artificial Intelligence to analyze RWE from electronic health records to identify patients with a high-risk trajectory and to validate a sensitive surrogate endpoint like eGFR slope for this specific population. Next, these patient characteristics would inform an ISCT, where a ‘virtual cohort’ is used to simulate the trial and computationally optimize its design, refining the inclusion criteria and sample size before enrolling a single patient. The trial would then be launched as an adaptive platform trial. As it runs, AI models would analyze incoming data from both clinic visits and DHTs (e.g., remote activity monitors) to predict which patient subgroups are responding best. This real-time analysis feeds directly into the adaptive protocol, allowing the trial to seamlessly enrich the population with predicted responders, thereby boosting efficiency and shortening the timeline. In this example, the synergy between methodological design (adaptive trial, surrogates) and computational tools (AI, RWE, ISCTs) creates a faster, more precise, and more powerful research engine than any single innovation could achieve alone.

## 5. Navigating Implementation: Overcoming Barriers in Innovation

Realizing the transformative potential of the methodological and computational innovations outlined previously requires confronting significant implementation hurdles [40]. A foundational barrier, particularly impacting AI applications and the generation of reliable RWE or the conduct of pragmatic trials integrated with routine care, lies in data quality, access, and standardization [14,40]. Healthcare data are often fragmented across disparate systems, incomplete, stored in non-standardized formats (especially unstructured text), and difficult to access due to privacy constraints and institutional silos. The performance and reliability of AI models are directly dependent on the quality and representativeness of the underlying training data. Poor quality data can lead to inaccurate or unreliable models. Similarly, the validity of RWE derived from sources like EHRs or claims databases hinges on data accuracy and completeness [15]. Furthermore, a critical concern is the potential for algorithmic bias, where AI models trained on data reflecting historical inequities in care access or diagnosis may perpetuate or even amplify those disparities, impacting fairness and generalizability. Ensuring robust data privacy and security in compliance with regulations (e.g., HIPAA, GDPR) is also paramount when handling sensitive patient information for these advanced analyses. Overcoming these data challenges requires concerted efforts in developing robust data governance frameworks, promoting interoperability and the use of common data models [61], investing in data quality improvement initiatives, and employing techniques to mitigate bias [40]. Establishing credibility and ensuring rigorous validation is essential for both computational tools and novel methodologies [42]. For AI models, particularly complex ‘black box’ algorithms like deep neural networks, demonstrating transparency and explainability is crucial for building clinical trust and facilitating error detection. Rigorous external validation across diverse populations and settings is necessary for ensuring models generalize beyond their training data, yet such validation is often lacking [40]. For ISCTs, establishing model credibility for a specific Context of Use (COU) is the central challenge [41]. This demands meticulous Verification (confirming correct implementation), Validation (comparing model predictions against real-world data), and Uncertainty Quantification (VVUQ) [30,50]. The lack of widely accepted standards hinders consistency; thus, establishing Good Simulation Practices (GSPs), analogous to GCP/GLP, is vital for promoting harmonization and ensuring quality [49]. The regulatory landscape for these innovations is actively evolving. While agencies like the FDA and EMA encourage Model-Informed Drug Development (MIDD), clear and harmonized pathways for the acceptance of evidence derived from complex ISCTs or AI algorithms as primary support for regulatory decisions are still developing [42]. Navigating this requires early and ongoing engagement between researchers, sponsors, and regulators, transparent reporting, and building confidence through consistent, high-quality evidence generation. Legislative support, such as the FDA Modernization Act 2.0 allowing in silico alternatives to animal testing, signals progress. Ethical considerations are paramount and cut across all innovations [21]. Beyond managing bias and privacy, ensuring truly informed consent is challenging when participants must understand complex adaptive designs or the role of AI [62]. Accountability and liability in case of errors involving AI or simulation outputs need clear frameworks. There is an ethical imperative for ensuring equitable access to the benefits of these technologies and maintaining patient trust and the centrality of the human element in clinical care [21,62]. Specific ethical dilemmas arise in long-term trials regarding placebo use when effective therapies exist and ensuring surrogate endpoints genuinely reflect patient-meaningful benefit [1]. Ensuring equitable inclusion of underrepresented populations in trials employing these new methods also remains a critical ethical mandate requiring proactive strategies. Finally, successful adoption hinges on workforce development and practical integration. Effectively developing, validating, and utilizing AI and CM&S tools requires specialized expertise in data science, computation, and statistics, coupled with deep clinical domain knowledge [29,30]. Fostering interdisciplinary collaboration and investing in training programs for the nephrology research workforce are crucial. Furthermore, for tools to be impactful, they must integrate seamlessly into existing clinical workflows and EHR systems without imposing undue burden on clinicians. Even when robust evidence exists for an innovation (methodological or therapeutic), bridging the implementation gap to ensure consistent uptake in routine practice remains a persistent challenge, influenced by system-level factors, resource constraints, and clinician behavior [1,63]. The imperative to overcome these barriers is amplified by the context of the broader AI revolution currently transforming medicine. This is not a theoretical future; landmark achievements are already signaling a paradigm shift from AI as an auxiliary tool to a core engine of biomedical discovery. For instance, AI models like AlphaFold are reshaping molecular biology and drug discovery with highly accurate protein structure predictions [64], while other AI-driven analyses of clinical images are providing profound predictive value for the personalized diagnosis and treatment of patients [65]. Placing nephrology’s challenges within this powerful context allows us to frame the key questions for the next phase of research, moving from simply adapting existing technologies to actively leading innovation. A primary challenge will be to determine how general AI breakthroughs, such as large language models, can be effectively fine-tuned on the smaller, highly specialized datasets common in nephrology, like digitized biopsy slides or rare disease registries, to overcome data scarcity without perpetuating bias. Furthermore, a parallel effort must focus on creating the robust validation standards and “Good Simulation Practices” needed for a trustworthy AI framework specifically tailored for nephrology, which is essential for ensuring the transparency required for regulatory and clinical acceptance. Finally, to maximize our resources and accelerate progress across the specialty, a crucial avenue of research will be the development of frameworks that allow for the efficient validation and transfer of models across different types of kidney disease, for instance, adapting a model from diabetic kidney disease for use in glomerulonephritis [66]. Addressing this complex web of interconnected barriers through investment in infrastructure, development of standards, regulatory clarity, ethical oversight, workforce training, and robust validation is essential for unlocking the full potential of these innovations to improve clinical research in nephrology (Table 3).

## 6. Conclusions

Clinical research in nephrology stands at a critical juncture, poised for transformation. The persistent challenges of slow disease progression, patient heterogeneity, recruitment difficulties, and endpoint limitations have necessitated a departure from traditional trial paradigms. The adoption of adaptive designs, pragmatic methodologies, real-world evidence integration, validated surrogate endpoints, and personalized strategies offers significant improvements in trial efficiency, relevance, and patient-centricity. Crucially, the greatest potential lies not in the siloed application of these innovations, but in their synergistic integration. A future where AI, including advanced techniques like AR and CTGANs, analyzes rich real-world data (RWD) to inform adaptive platform trials simulated in silico, incorporating validated surrogates alongside digital endpoints derived via Digital Health Technologies (DHTs), represents a powerful vision for accelerating therapeutic development. Such integrated approaches promise trials that are faster, more cost-effective, more informative, and better tailored to individual patient needs. From the patient’s perspective, this integrated approach translates into tangible benefits. For example, the use of in silico trials and synthetic control arms generated by CTGANs can reduce the number of participants assigned to placebo, increasing their chance of receiving a potentially effective therapy. Augmented Reality guidance for procedures like renal biopsies can improve precision, which may reduce the need for painful repeat procedures. Furthermore, efficient trial designs powered by remote monitoring via DHTs not only decrease the burden of frequent travel to clinical sites but also accelerate the entire research lifecycle. This shortened timeline means that successful new treatments can move from discovery to the broader patient community sooner. By embracing methodological and computational innovation thoughtfully and responsibly, fostering collaboration, investing in infrastructure and training, and maintaining rigorous scientific and ethical standards, the nephrology community can usher in an era of more efficient, effective, and equitable clinical research. While our proposed integration model represents a novel conceptual framework, we acknowledge it requires empirical validation through pilot studies. This review synthesizes existing evidence to propose future directions rather than presenting validated clinical protocols. The synergistic approach outlined here should be viewed as a roadmap for future research validation. The ultimate goal is to accelerate the delivery of impactful therapies and improve the lives of millions worldwide affected by kidney disease.

## Figures and Tables

**Figure 1 sensors-25-04909-f001:**
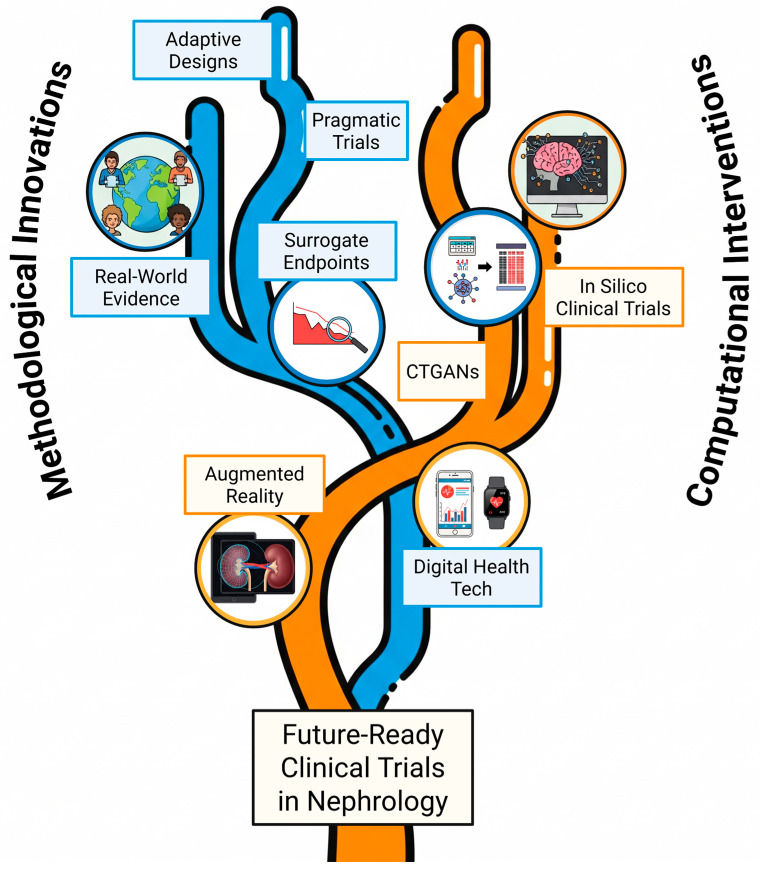
**The synergistic convergence of innovation in trial design.** The figure illustrates a paradigm shift in clinical research in nephrology, where two key streams of innovation converge. The left pathway represents methodological advancements, including adaptive designs, pragmatic trials, real-world evidence (RWE), and validated surrogate endpoints. The right pathway embodies computational power, driven by Artificial Intelligence (AI), in silico clinical trials (ISCTs), Augmented Reality (AR), and generative models. Their synergistic integration forms a unified approach, leading to future clinical trials that are more efficient, precise, and patient-centric. Created in BioRender. Strizzi, C. T. (2025) https://BioRender.com/b232ph0.

**Figure 2 sensors-25-04909-f002:**
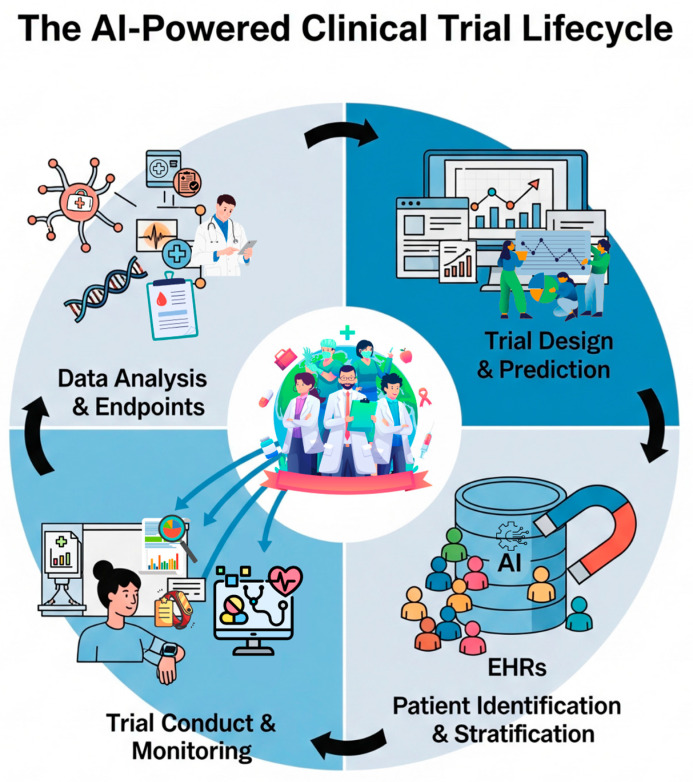
**The AI-powered clinical trial lifecycle.** Artificial Intelligence offers transformative potential across the entire clinical trial process, as depicted in this circular workflow. The cycle begins with Trial Design and Prediction, where AI analyzes historical data to optimize protocols. In Patient Identification and Stratification, machine learning screens electronic health records (EHRs) to accelerate recruitment. During Trial Conduct and Monitoring, AI supports remote data collection from digital health technologies (DHTs) and helps predict non-adherence. Finally, in Data Analysis and Endpoint Development, AI uncovers complex patterns in multi-modal data and helps derive novel digital endpoints, generating deeper insights.

**Table 1 sensors-25-04909-t001:** Persistent challenges in traditional clinical trials in nephrology.

Challenge	Description	Impact on Clinical Trials
Slow Disease Progression	Many kidney diseases, especially CKD, progress slowly, leading to low event rates for hard clinical endpoints (e.g., ESKD, mortality).	Requires very large sample sizes and long follow-up durations, increasing costs and time.
Treatment Success Paradox	Effective new therapies (like SGLT2i) lower baseline risk in participants, making it harder to show incremental benefits of new agents.	Decreases the statistical power of traditionally designed trials, necessitating the enrollment of larger populations, extended trial durations, or focusing on high-risk cohorts.
Recruitment and Retention	Difficulty finding and keeping participants due to complex protocols, participant burden (time, cost, travel), lack of awareness, mistrust.	Jeopardizes trial feasibility, statistical power, timelines, and generalizability of results.
Lack of Diversity	Underrepresentation of minority ethnic and socioeconomically disadvantaged groups who often have a higher disease burden.	Limits generalizability and equity of research findings.
Disease Heterogeneity	Kidney diseases encompass diverse etiologies (DKD, GN, ADPKD, AKI) and varying progression rates.	Complicates trial design, patient selection, endpoint definition, and interpretation; a “one-size-fits-all” approach is often ineffective.
Endpoint Selection	Balancing the clinical relevance of hard endpoints against the feasibility challenges of low event rates.	Drives the need for validated surrogate endpoints (e.g., eGFR slope), but validation is rigorous and ongoing.

Abbreviations: ADPKD: autosomal dominant polycystic kidney disease, AKI: acute kidney injury, CKD: chronic kidney disease, DKD: diabetic kidney disease, ESKD: end-stage kidney disease, GN: glomerulonephritis, SGLT2i: sodium-glucose co-transporter 2 inhibitors.

**Table 2 sensors-25-04909-t002:** Methodological and computational innovations enhancing nephrology trial design.

Innovation	Description	Potential Benefit(s)
Adaptive Designs	Allow pre-planned modifications (e.g., sample size, arm dropping) based on interim data, often using Bayesian methods.	Increased efficiency, flexibility, shorter duration, smaller sample size, ethical advantages (stopping early).
Platform Trials	Test multiple interventions against a common control group using a master protocol.	Dramatically increased efficiency for evaluating numerous therapies, especially for heterogeneous diseases.
Pragmatic Clinical Trials (PCTs)	Evaluate interventions in real-world settings with broad eligibility, often using routine data collection (EHRs).	Increased generalizability, relevance to routine care, potentially lower cost.
Real-World Evidence (RWE)	Leverage data from EHRs, registries, claims databases to understand long-term effectiveness/safety in diverse populations.	Complements RCT data, provides insights into routine practice, supports regulatory decisions in some contexts.
Digital Health Technologies (DHTs)	Use wearables, sensors, apps for remote monitoring (e.g., BP, weight, ePROs).	Reduced participant burden, wider participation, frequent data collection, potential for novel/sensitive digital endpoints.
Surrogate Endpoints (Validated)	Use markers (e.g., % eGFR decline, eGFR slope) reliably predicting clinical outcomes to shorten trials.	Allows for smaller/shorter trials, feasible when hard endpoints are rare, accelerates development.
Conditional Tabular Generative Adversarial Networks (CTGANs)	AI-driven technique engineered to learn the distributions within real-world tabular data and generate high-fidelity, synthetic patient records.	Augments scarce datasets in rare diseases, can improve the robustness of predictive models, helps preserve data privacy, and enables more reliable subgroup analyses.
Augmented Reality (AR)	Overlays real-time, 3D imaging data (e.g., CT scans) onto a clinician’s view of the operative field to provide intuitive guidance during procedures.	Improves procedural precision and efficiency for interventions like renal biopsies and vascular access cannulation, reduces inter-operator variability, and contributes to higher quality endpoint data.

Abbreviations: BP: blood pressure, CT: computed tomography, eGFR: estimated glomerular filtration rate, EHRs: electronic health records, ePROs: electronic reported outcomes, RCT: randomized controlled trial.

**Table 3 sensors-25-04909-t003:** Implementation barriers for innovative trial designs.

Barrier Category	Description	Implications
Data	Quality, fragmentation, lack of standardization, access issues, privacy.	Hinders reliable AI model training, RWE generation, PCT integration; requires robust governance and interoperability efforts.
	Algorithmic bias.	AI models trained on biased data may perpetuate/amplify health disparities; requires mitigation techniques.
Validation	Establishing AI model credibility (transparency, explainability, external validation).	Crucial for clinical trust and regulatory acceptance; often lacking rigorous external validation.
	Establishing ISCT model credibility (VVUQ—Verification, Validation, Uncertainty Quantification).	Central challenge requiring meticulous processes and standards (e.g., Good Simulation Practices) for regulatory trust.
Regulatory	Evolving landscape, lack of harmonized pathways for AI/ISCT evidence.	Creates uncertainty; requires early engagement with regulators and transparent reporting.
Ethical	Informed consent complexity (for adaptive designs, AI use).	Requires clear communication to ensure participants understand the trial processes.
	Bias, privacy, accountability, equity, maintaining patient trust.	Foundational concerns requiring careful oversight, ethical frameworks, and focus on equitable benefit distribution.
Workforce	Need for specialized expertise (data science, computation) and interdisciplinary collaboration.	Requires investment in training and fostering collaboration between clinical and computational experts.
Integration	Seamless integration into clinical workflows and EHRs.	Essential for practical adoption; tools should not unduly burden clinicians.
Adoption	Bridging the implementation gap for proven innovations.	Requires addressing system-level factors, resource constraints, and clinician behavior beyond just evidence generation.

Abbreviations: AI: artificial intelligence, EHRs: electronic health records, ISCT: in silico clinical trial, PCT: pragmatic clinical trial, RWE: real world evidence.

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
