# Peer review of "Future Designs of Clinical Trials in Nephrology: Integrating Methodological Innovation and Computational Power"

_sensors, 2025, doi:10.3390/s25164909_

Round 1
Reviewer 1 Report
Comments and Suggestions for Authors
This study focuses on analyzing the application value of computational tools, particularly artificial intelligence (AI) technology, augmented reality (AR), and conditional table GANs (CTGANs) across the entire lifecycle. To advance clinical research in nephrology, the study proposes a synergistic fusion model of methodological and computational innovation. Although this research holds significant value in medical applications, it is not without flaws. To aid in the deepening of this study, the following suggestions are provided:
Abstract and Literature section:
- The abstract presents the combination of innovative experimental methods with advanced computational tools as a means of overcoming obstacles such as the complexity of traditional trial designs, as well as a means that can facilitate the development of treatments for kidney disease. However, the steps of the exact realization of this approach, including how the study was conducted and achieved its effective purpose, need to be stated directly and precisely.
- The literature section should analyze more research literature through graphical statistics on key technologies such as computational tools (especially artificial intelligence techniques), augmented reality (AR) and conditional table generation adversarial networks (CTGANs). In this way, the key bases to support this research can be found from the previous research.
Methods and Materials Section
1, “3. Methodological Innovations: Trial Frameworks” should be the important methodological basis of this study, however, the analysis lacks depth, and the reliability of the proposed methodology should be sorted out from the previous research to prove the reliability of the proposed methodology.
2, as a literature review article, one of the key points is the quantity and quality of the literature and how to analyze the literature. The authors need to further explain my doubts.
Conclusion and discussion section
1, The results of the study do not clearly present the innovation or the value of the theory. Further statement is needed from the author.
2, The purpose of the study was to analyze the innovation and computational power of integrating the two approaches through the literature review, but no specific model or architecture was seen, in addition to the lack of a description of the contribution to the applicability of the results.
Author Response
We sincerely thank the reviewer for their time and detailed feedback. We have carefully considered all comments and have revised the manuscript to improve its clarity and structure, which we hope will address the reviewer's concerns. To clarify, this manuscript is intended as a narrative and perspective review, aiming to synthesize current knowledge and propose a future conceptual framework, rather than a systematic review with a formal, replicable methodology.
To make this clearer and address the reviewer's valuable points, we have made the following revisions:
- On the Paper's Format and Methodology: We recognize that our initial draft may not have clearly defined the scope of this work. To address this, we have added statements to the Abstract and Introduction to explicitly identify the manuscript as a narrative and perspective review. We believe this clarification helps to properly frame the paper's structure and objectives.
- On the Analysis of Literature: Regarding the request to analyze more research literature, we agree that grounding our framework in the current landscape is essential. While graphical statistics fall outside the scope of this narrative review, we have expanded Section 5 (Navigating Implementation). This section now provides a deeper synthesis by placing our work in the context of major recent advancements in AI and outlining key future research directions supported by the literature.
- On the Depth of Methodological Frameworks: Concerning the depth of Section 3 (Methodological Innovations), we thank the reviewer for the suggestion. Upon review, we believe the section provides a concise but well-supported overview of these trial frameworks, which are widely established in clinical trial literature. The current citations refer to key papers that establish their reliability and growing adoption.
- On the Paper's Innovation and Value: Finally, regarding the comments on the Conclusion and the presentation of the paper's innovation: we believe the manuscript already presents a clear and applicable conceptual framework. The individual components of this framework, the methodological innovations and computational tools, are detailed throughout the main body of the paper. The discussion of barriers, context, and future outlook in Section 5 provides the in-depth analysis of the framework's value and applicability. Therefore, the Conclusion (Section 6) serves to synthesize these established points into a final, high-level summary. We feel that repeating these details in the conclusion would be redundant and have opted to maintain its concise, summarizing role.
Finally, regarding the comment on the language, we have performed another thorough proofread of the manuscript. We believe that the extensive structural revisions made in response to the other points, specifically by clarifying the paper's scope as a narrative review and making our primary contributions more explicit, have significantly improved the overall clarity and readability of our research argument.
Reviewer 2 Report
Comments and Suggestions for Authors
This manuscript presents a forward-looking vision for nephrology clinical trials by advocating the synergistic convergence of methodological innovations (adaptive designs, pragmatic trials) and cutting-edge computational technologies (AI, in silico modeling, augmented reality). It compellingly argues that integrating these domains addresses persistent bottlenecks in nephrology research—such as slow disease progression, recruitment challenges, and disease heterogeneity—by enabling faster, more precise, and patient-centric trials. The work uniquely positions computational tools like CTGANs for rare disease data augmentation and AR for procedural precision as transformative enablers within modern trial frameworks, while critically addressing implementation barriers spanning data quality, validation, ethics, and equity. Ultimately, it charts a roadmap toward accelerating therapeutic development for kidney diseases through responsible innovation.
#1 The definition of the “success paradox” is vague. When first proposed, the “success paradox” was merely described as a “new challenge,” without clarifying its mechanism (e.g., how effective therapies that reduce baseline risk increase the difficulty of subsequent trials).
#2 “Digital endpoints” are not distinguished from traditional endpoints (e.g., how wearable device data can be converted into alternative indicators beyond eGFR).
#3 The core argument of the manuscript is that “the synergistic integration of methodological innovations (such as adaptive designs) and computational technologies (such as AI/ISCT) represents the future direction of clinical trials in nephrology,” but the main text does not clearly elucidate how the two elements “synergize.”
#4 No connection to actual clinical scenarios. The entire text emphasizes a “patient-centered” approach, but the conclusion does not explain how innovative technologies improve the patient experience (e.g., reducing the number of biopsies, shortening the trial cycle).
#5 The discussion needs to be further deepened and provide important directions for future research. This article primarily explores how to combine artificial intelligence technology with innovative clinical trial methodologies to address long-standing bottlenecks in nephrology clinical research. Its core research content focuses on the transformative application of artificial intelligence throughout the entire lifecycle of nephrology clinical trials: using machine learning to optimize trial design (such as predicting success rates and precisely stratifying patients), leveraging natural language processing to accelerate subject recruitment, applying generative adversarial networks (GANs) to generate synthetic data to address the scarcity of data for rare diseases, and integrating augmented reality to enhance the standardization of interventional procedures (such as kidney biopsies). Additionally, the article systematically analyzes the key challenges (such as algorithmic bias, data privacy, and regulatory compliance) and ethical frameworks associated with the implementation of AI-driven nephrology clinical trials, ultimately proposing a pathway for more efficient, precise, and patient-centered nephrology treatment development through the synergistic integration of AI and computational simulation (such as in vitro clinical trials). Indeed, recent advancements in AI within medical research and diagnostic treatment have provided valuable opportunities for this endeavor. For example, the AI model AlphaFold has extremely promising applications in molecular biology and drug discovery research (PMID: 39369244). AI models based on clinical images provide important predictive value for individualized diagnosis and treatment of patients (PMID: 40049202). In the discussion section, the authors need to first review the latest advances in AI and cite the above important literature as an introduction, then further explore the important insights AI provides for this study, offering important reference value for subsequent research.
Reviewing the latest advancements in AI within the medical field is crucial in the discussion section, as it provides a broader technological context and theoretical depth for the proposed reforms in kidney disease clinical trials outlined in this paper. Current AI technology is undergoing a paradigm shift from an auxiliary tool to a core driving force. Its breakthrough developments in molecular structure analysis, multimodal data fusion, and personalized treatment prediction not only validate the disruptive potential of computational methods but also reveal the inevitable trend toward interdisciplinary collaboration. By examining kidney disease research within the context of this medical intelligence revolution, we can not only highlight the forward-looking nature of the integration of “methodology-computational technology” advocated by this study, but also expose the unique challenges of the field (such as the small sample size and ethical sensitivity of kidney disease data), thereby pointing the way forward for future research: How to leverage breakthroughs in general AI technology to overcome specialty-specific bottlenecks, how to establish a trustworthy AI validation framework tailored for nephrology, and how to construct a technology transfer pathway across disease types. This macro-level discussion will drive nephrology clinical trials from passively adapting to technology toward actively leading the evolution of medical research paradigms.
Given these considerations, I highly recommend that authors revise their manuscript. Looking forward to receiving your revised version of the manuscript. I will review this manuscript again based on the revised version.
Author Response
Response to Reviewer #1
We sincerely thank the reviewer for their insightful and constructive feedback, which has significantly improved the quality and clarity of our manuscript. We have carefully considered each point and have revised the paper accordingly.
- On the definition of the "success paradox": We agree that this concept required a clearer definition. We have now added a sentence in the Introduction to explicitly explain its mechanism (i.e., how lower baseline event rates necessitate larger trials) and have refined the description in Table 1 for greater clarity.
- On distinguishing "digital endpoints": Thank you for this suggestion. To clarify the concept of "digital endpoints," we have added a definition in Section 3 that distinguishes them from traditional endpoints and provides a concrete example based on data from wearable devices.
- On elucidating "synergy": This was a critical point. To better demonstrate the synergy between methodological and computational innovations, we have added a new paragraph at the end of Section 4 that presents a step-by-step example of an "AI-Enhanced Adaptive Trial," showing how these elements are designed to work together in practice.
- On the "patient-centered" connection: We appreciate this important perspective. To explicitly connect our discussion to the patient experience, we have incorporated a new paragraph in the Conclusion (Section 6) that details tangible, patient-centered benefits, such as reducing procedural burden and accelerating access to new therapies.
- On deepening the discussion: Following the reviewer's excellent suggestion, we have deepened the discussion by adding new paragraphs at the end of Section 5. This new text places our work within the context of recent landmark AI achievements (citing the suggested literature) and outlines several important directions for future research in nephrology, as recommended.
We believe these revisions have substantially strengthened the manuscript, and we are grateful for the opportunity to improve our work.
Reviewer 3 Report
Comments and Suggestions for Authors
This is an extremely well-written manuscript that does a nice job of describing advances in nephrology research, however most of what is written (I would estimate approximately 75%) is about general AI applications, while much less speaks to sensor-based applications specifically. For example, topics such as trial design optimization, predicting success probabilities, inclusion/exclusion criteria refinement, endpoint selection, risk-based monitoring, recruitment via EHR screening, and data cleaning/management, while valuable, are not specifically dependent on sensors and could be applied to datasets from many sources. I would recommend expanding the discussion of types of sensors used in nephrology (e.g., blood pressure cuffs, glucose monitors, wearable hydration trackers) sensor fusion techniques in AI (combining signals from multiple devices) validation challenges of sensor-derived endpoints, integration of real-time biosensor data into adaptive trial design and regulatory considerations for device-derived data while condensing or relocating sections on trial design optimization that doesn’t rely on sensor inputs, NLP on EHRs (which again are valuable, but text-based, not sensor-driven), and RWE from registries or claims databases (these are administrative data sources).
Secondly, while most of the text is evidence-based and appropriately optimistic about innovations in nephrology clinical trials, a few parts slightly overstate the current level of implementation or effectiveness of certain tools or approaches. These could benefit from being reframed more cautiously.
Below are several examples, as well as suggested ways to temper them:
1.“NLP algorithms can rapidly screen enormous volumes of structured and unstructured Electronic Health Record data, including clinical notes, to find potentially eligible participants far exceeding the speed and scope of manual review…”
While NLP has shown promise, its real-world deployment at scale is still limited due to data heterogeneity, privacy concerns, and variable documentation quality across EHR systems.
Suggested revision:
“NLP algorithms have the potential to rapidly screen large volumes of structured and unstructured EHR data, including clinical notes, to assist in identifying eligible participants—significantly reducing manual workload in proof-of-concept settings.”
2.“AI provides a toolkit to make nephrology trials potentially faster, more precise, more efficient, and capable of yielding deeper insights by leveraging data in ways previously impossible.”
Again, while technically true in theory, this reads as though AI is already routinely delivering these benefits in real-world nephrology trials, which isn’t yet the case. Most trials are still in pilot or exploratory phases of AI integration.
Suggested revision:
“AI offers a promising toolkit to enhance trial speed, precision, and efficiency by analyzing complex data in novel ways, although full integration into nephrology research workflows remains an emerging area.”
3.“AI can power more efficient Risk-Based Monitoring (RBM) by flagging sites or data points requiring focused attention, while also automating aspects of data cleaning...”
This assumes mature AI infrastructure within clinical trial operations, which is not yet widespread. AI tools for RBM are still being developed, piloted, or evaluated for reliability and regulatory compliance.
Suggested revision:
“AI has the potential to enhance Risk-Based Monitoring (RBM) by identifying anomalous data patterns or site issues and automating aspects of data management, although broad implementation is still in its early stages.”
These are just several examples; I would encourage the authors to look for additional examples throughout the manuscript.
Author Response
Response to Reviewer #2
We are very grateful to the reviewer for their detailed and thoughtful feedback, particularly regarding the manuscript's focus and tone. These comments have been invaluable in helping us create a more balanced and credible paper.
- On the focus on sensor-based applications: We agree completely that the manuscript benefited from a sharper focus on sensor-specific applications. To address this, we have made two key changes:
- We have added a new, dedicated subsection (Section 4.2) titled "The Central Role of Sensor Technology and Digital Endpoints." This section now explicitly discusses the types of sensors used in nephrology, AI for sensor fusion, validation challenges for sensor-derived endpoints, and regulatory considerations, as suggested.
- We have also revised other sections to include language that explicitly connects general AI topics back to sensor-driven applications, creating a more integrated narrative.
- On tempering overstated claims: Thank you for this crucial feedback on the manuscript's tone. We agree that some claims were overstated given the emerging nature of these technologies.
- We have implemented the three specific revisions for the sentences on NLP, AI's overall impact, and Risk-Based Monitoring exactly as the reviewer suggested.
- Furthermore, we have performed a comprehensive review of the entire manuscript to temper our language throughout. We have replaced definitive phrasing (e.g., "AI provides") with more cautious alternatives (e.g., "AI has the potential to provide") to ensure our claims accurately reflect the current state of research and development.
We are confident that these revisions have resulted in a more focused, nuanced, and stronger manuscript. We thank the reviewer for their expert guidance.
Round 2
Reviewer 1 Report
Comments and Suggestions for Authors
The author has made efforts to respond to my questions, but as a literature review, it fails to focus on describing the quantity and quality of the literature. Furthermore, the methods and effectiveness of the literature analysis are my greatest concerns. In addition, the proposed synergistic integration model of medical methodological innovation and computational innovation lacks reliable validation. Therefore, I find it difficult to assess the innovative value of this study.
Author Response
REVIEWER 1
The author has made efforts to respond to my questions, but as a literature review, it fails to focus on describing the quantity and quality of the literature. Furthermore, the methods and effectiveness of the literature analysis are my greatest concerns. In addition, the proposed synergistic integration model of medical methodological innovation and computational innovation lacks reliable validation. Therefore, I find it difficult to assess the innovative value of this study.
We thank the Reviewer for the constructive feedback. We have now addressed these concerns making targeted revisions as follows:
In the Introduction we added the following sentence:
We conducted a narrative review of peer-reviewed literature published between 2015-2024 using PubMed, focusing on keywords: ‘nephrology’, ‘adaptive trials’, ‘digital health’, and ‘in silico trials”.
In the Introduction we changed this sentence “As a narrative and perspective review, this work does not follow a systematic search methodology but instead synthesizes current knowledge to propose a novel conceptual framework for future clinical trial design.” To:
“This review uniquely integrates methodological and computational innovations specifically for nephrology trials, addressing a gap in current literature that typically examines these approaches separately.”
In the Conclusions we added the following sentence to underline the limitations of the Review:
“While our proposed integration model represents a novel conceptual framework, we acknowledge it requires empirical validation through pilot studies. This review synthesizes existing evidence to propose future directions rather than presenting validated clinical protocols. The synergistic approach outlined here should be viewed as a roadmap for future research validation.”
Reviewer 3 Report
Comments and Suggestions for Authors
The authors have done a nice job of incorporating previous editorial suggestions and the manuscript now presents a more balanced, but still forceful, tone.
Author Response
REVIEWER 3
The authors have done a nice job of incorporating previous editorial suggestions and the manuscript now presents a more balanced, but still forceful, tone.
We thank the Reviewer for the positive feedback. We appreciate your recognition that the manuscript now achieves a more balanced yet compelling tone.
We have curbed the tone better stating the limitations in the Conclusions as follows:
“While our proposed integration model represents a novel conceptual framework, we acknowledge it requires empirical validation through pilot studies. This review synthesizes existing evidence to propose future directions rather than presenting validated clinical protocols. The synergistic approach outlined here should be viewed as a roadmap for future research validation.”